# Immediate Patient Satisfaction with Dental Esthetics After Endodontic and Prosthodontic Treatment of Dental Dyschromia

**DOI:** 10.3390/dj13010044

**Published:** 2025-01-20

**Authors:** Adrian Jantea, Sorin Gheorghe Mihali, Dan Lolos, Anca Tudor, Roxana Oancea, Liliana Porojan

**Affiliations:** 1Faculty of Dental Medicine, “Victor Babeș” University of Medicine and Pharmacy Timișoara, Eftimie Murgu Square No. 2, 300041 Timișoara, Romania; jantea.adrian@umft.ro (A.J.); lolosdan@umft.ro (D.L.); 2Department of Prosthodontics, Faculty of Dentistry, “Vasile Goldis” Western University of Arad, 94 Revolutiei Blvd., 310025 Arad, Romania; 3Medical Informatics and Biostatistics, Research Center in Dental Medicine Using Conventional and Alternative Technologies, Faculty of Dental Medicine, “Victor Babes” University of Medicine and Pharmacy, 2 Eftimie Murgu Square, 300041 Timisoara, Romania; atudor@umft.ro; 4Department of Preventive and Community Dentistry, Faculty of Dental Medicine, “Victor Babeş” University of Medicine and Pharmacy, 300041 Timişoara, Romania; roancea@umft.ro; 5Translational and Experimental Clinical Research Centre in Oral Health, Department of Preventive, Community Dentistry and Oral Health, University of Medicine and Pharmacy “Victor Babes”, 300040 Timisoara, Romania; 6Center for Advanced Technologies in Dental Prosthodontics, Department of Dental Prostheses Technology (Dental Technology), Faculty of Dental Medicine, “Victor Babeș” University of Medicine and Pharmacy Timișoara, Eftimie Murgu Square No. 2, 300041 Timișoara, Romania; liliana.porojan@umft.ro

**Keywords:** patients’ satisfaction, prosthodontic treatment, endodontic treatment, dyschromia, devitalized teeth, ceramic materials

## Abstract

**Objectives:** This study aimed to evaluate patients’ satisfaction with the esthetic outcomes of combined endodontic and prosthetic treatments for devitalized or dyschromic teeth, a condition influenced by various intrinsic and extrinsic factors that present a growing concern in modern dentistry. **Methods:** A total of 104 patients, including 43 men and 61 women, underwent treatment using lithium disilicate restorations for esthetic zones and zirconium oxide restorations for regions with higher occlusal demands. Patient satisfaction was evaluated through a post-treatment questionnaire, classifying responses as either “satisfied” or “dissatisfied”. Dissatisfied participants were further asked to specify their concerns. **Results:** The study revealed a high satisfaction rate of 93%. Dissatisfaction was slightly more prevalent among women than men, but this difference was statistically insignificant. The primary reasons for dissatisfaction included darker restoration color, chipping, and gingival recessions. **Conclusions:** Combined endodontic and prosthetic treatments, utilizing lithium disilicate and zirconium oxide restorations, achieved high levels of patient satisfaction. Nevertheless, addressing specific issues, such as color matching and gingival health, could enhance outcomes further.

## 1. Introduction

When we refer to a successful prosthetic restoration today, it is not enough to consider only the survival rate of restorations after performing prosthetic and endodontic treatments. Regarding tooth discoloration today, due to the influence and effect that social media has on the perception of prosthetic treatments [1], patients’ expectations have increased considerably [2,3].

Substantial increases in esthetic requirements nowadays have determined that a simple dental discoloration can have a negative impact on the individual’s mental health [4]. In today’s context, the color of teeth plays a significant role in the perception of esthetics, especially given the considerable interest patients show in private practices regarding teeth whitening treatments [5]. Apart from the mechanical properties necessary for successful prosthetic restorations, we have to accept that achieving satisfactory esthetics is equally vital by today’s standards. In order to facilitate these standards of adequate esthetic treatments, we will have to corroborate the endodontic treatment with the prosthetic one. The color of the remaining tooth after its preparation or reconstruction can influence the color of the final restoration, especially in patients with devitalized teeth or dental dyschromia.

Changes to the structures comprising the coronal portion (enamel, dentin, and dental pulp) can lead to alterations in the outward color appearance of the tooth, influenced by its light transmission and reflection properties [6]. The color of the tooth, as perceived visually, relies on the quality of the light that is reflected from it and is directly related to the amount of incident light it receives. Color changes in teeth, particularly dyschromia, occur due to alterations in the tooth’s internal and external structures, which affect its light reflection and transmission properties. Intrinsic factors such as metabolic diseases, genetic conditions, trauma, or the use of certain medications (e.g., tetracycline) can change the tooth’s internal composition—specifically the dentin and enamel—resulting in discoloration that originates from within the tooth. This happens because the internal structure of the tooth becomes opaquer or stained, altering how light interacts with the tooth. On the other hand, extrinsic factors, such as dietary habits, smoking, and the consumption of staining beverages like coffee and wine, deposit pigments on the tooth surface, which can cause discoloration without changing the internal structure. The “why” behind these changes is often linked to how the tooth’s surface and internal layers interact with light, affecting the way it appears to the observer. Furthermore, degradation or wear of dental materials used in restorations (such as crowns or fillings) can also cause visible color discrepancies over time.

Dental dyschromia is classified according to the location of the stain, i.e., intrinsic or extrinsic. Intrinsic dyschromia develops when there is a modification in the structural composition or thickness of the dental hard tissues. Certain systemic factors or some metabolic diseases are incriminated when it comes to the damage caused in the development of the dentition that can cause discoloration as a consequence. In addition to local factors, such as injury or trauma, intrinsic discolorations can occur due to: amelogenesis imperfecta, dentinogenesis imperfecta, fluorosis, pulpal hemorrhagic products, root resorption, aging, enamel hypoplasia, congenital hyperbilirubinemia, congenital erythropoietic porphyria, alkaptonuria, and tetracycline staining.

The primary reasons for intrinsic pre-eruptive dyschromia often stem from tetracycline ingestion during odontogenesis [7,8,9,10,11,12]. Nonetheless, less common instances involve congenital enamel/dentine defects or hematological disorders. Extrinsic discolorations are categorized based on their source, either metallic or non-metallic in origin [6,13].

Dyschromias resulting from intrinsic and extrinsic factors as well as from failed endodontic treatments can be corrected using both prosthetic and chemical methods.

We will focus solely on mentioning chemical methods, specifically chemical bleaching, inside/outside bleaching or walking bleach [14,15,16]. These techniques involve applying hydrogen peroxide to the crown of the tooth, either directly into the endodontic cavity or indirectly through a chemical reaction involving carbamide or sodium peroxide.

In our study, we refrained from employing any tooth bleaching methods. We believe that these techniques contain components that could potentially cause significant structural changes to human dental enamel due to the articles that support this aspect in the specialized literature [17,18,19,20,21,22].

After performing an appropriate endodontic treatment, the clinicians will have to analyze the color of the dental preparation, the reconstruction performed, or the remaining tooth. Due to the fact that the color of the substrate can influence the color of the overall restoration, we applied either more invasive preparation concepts, or we used a certain appropriate ceramic mass, or the reconstruction of the abutment was carried out in such a way so as to not affect the final color. With these aspects taken into account, we also relied on obtaining the desired color by removing 0.2 mm to 0.3 mm of dental hard tissue for each shade [23].

In this study, we utilized lithium disilicate and zirconium oxide restorations to effectively restore devitalized and dyschromic teeth.

In most cases, considering the limited esthetic properties of zirconium oxide restorations in the frontal area [24], we opted for lithium disilicate restorations based on their esthetic properties and biocompatibility [25,26], taking into account the specific clinical cases.

For clinical situations involving dental units exposed to higher occlusal demands, we opted for zirconium oxide restorations due to their enhanced structural resistance properties [27,28,29].

The provisional restoration plays a crucial role in the success of endodontic treatment, as it must provide an optimal marginal seal to prevent bacterial contamination of the endodontically treated tooth until the final restoration is fixed [30]. The significance of both the final and provisional restoration lies in their ability to achieve optimal marginal closure, thereby preventing bacterial contamination of the treated tooth [31,32]. The crown should establish an ideal and enduring marginal closure with the tooth, safeguarding the remaining hard tissues while simultaneously restoring its function and natural morphology. To maintain the tooth within the dental arch for an extended period, it is essential to perform the endodontic treatment accurately, adhering strictly to all the steps of mechanical and antiseptic procedures [33,34,35]. The complete eradication of all microorganisms and the residual dental pulp is imperative, followed by a thorough disinfection of the root canals [36,37]. The quality of both the canal obturation and the build-up are equally vital factors to consider [38]. Whether or not to use corono-radicular devices plays a crucial role in achieving the strength of the treated tooth. When significant coronal destruction is present, the use of one or more corono-radicular devices becomes necessary. These devices provide support to the reconstruction material and the tooth, thereby preventing its potential loss in the future [38,39,40,41,42].

Presently, an increasing number of patients prefer all-ceramic restorations over metal or metal–ceramic options, seeking improved esthetics [43].

The translucency difference between zirconia and lithium disilicate ceramics is due to their structural compositions. Zirconia, being dense, is durable but lacks translucency, making it ideal for posterior restorations. In contrast, lithium disilicate, a glass-ceramic, is more translucent and better suited for anterior restorations due to its natural light-reflecting properties. When comparing all-ceramic to metal–ceramic restorations, esthetics, biocompatibility, and corrosion resistance are key. All-ceramic restorations are biocompatible and corrosion-resistant, offering better esthetics. Metal–ceramic restorations are durable but may have issues with corrosion and limited esthetics. The metal infrastructure in metal–ceramic restorations provides strength but reduces translucency, affecting esthetics, particularly in the anterior region. Teeth change color after endodontic treatment due to the loss of blood supply, which reduces light reflection, leading to a darker appearance. Additionally, root canal materials can contribute to discoloration.

On many occasions, an endodontically treated tooth presents complications due to inadequate restoration, even reaching its loss not so much because of the failure of the root canal treatment as because of the difficulty of the restoration. Depending on the area where the tooth is located, but also on the degree of dyschromia it has, a certain class of all-ceramic materials have been used. When dealing with significant dyschromia that reaches the limits of the dental preparation, different concepts will have to be used to allow the technician to mask it and thus manage to satisfy the patient’s esthetic requirements.

This study aims to evaluate patient satisfaction concerning esthetic outcomes following endodontic and prosthetic treatments for devitalized teeth or teeth with dental dyschromia. The study was conducted over the course of one year, with follow-up data collected at five years for additional insights into long-term outcomes.

## 2. Materials and Methods

This study was meticulously designed and executed in full accordance with the ethical principles delineated in the Declaration of Helsinki. Formal approval was obtained from the Ethics Committee of the University of Medicine and Pharmacy “Victor Babeș”, Timișoara, Romania (Nr.65/03.04.2023 rev 2024). Stringent measures were implemented to safeguard the welfare and confidentiality of the participants throughout the research process.

The study enrolled a cohort of 105 patients undergoing prosthetic dental treatments, each presenting with dyschromia in at least one dental unit. Gender distribution among the participants comprised 62 women and 43 men.

Inclusion criteria were specifically defined to ensure a consistent and relevant study population. Patients exhibiting the dyschromia of their teeth, including discoloration of vital teeth induced by medication (such as intrauterine administration of tetracycline) and discoloration of devitalized teeth resulting from secondary caries, inadequate endodontic treatments, or restorative interventions performed without proper rubber dam isolation or flawed conceptual approaches, were included in the study (Figure 1, Figure 2 and Figure 3).

Exclusion criteria were also established to maintain the integrity of the study. Patients with systemic conditions affecting oral health, those undergoing concurrent orthodontic treatment, and individuals with a history of allergy to the dental materials used in this study were excluded. All treatments were performed in appropriate clinical settings.

After the treatment was performed participants were administered a structured questionnaire (see Appendix A), presenting two response options: satisfaction or dissatisfaction following the discoloration dental treatment. Dissatisfied participants were further prompted to specify their reasons, which included issues such as chipping, darker final restoration color, and gingival recessions.

Data collected from the questionnaires were subjected to comprehensive statistical analysis using JASP v0.17.1 software to elucidate patterns, trends, and associations regarding treatment satisfaction and dissatisfaction among the study population.

In this research, dental restorations were performed using various materials, including lithium disilicate, zirconium oxide, and metal–ceramic. The choice of material was determined by both specific clinical indications and patient preferences, resulting in considerable variability in the restoration time.

The primary objective of our study was to evaluate patient satisfaction rates following dental restorations, irrespective of the time required to perform these restorations. Although we acknowledge that restoration time can influence patient satisfaction, we deemed it more crucial to focus on the overall satisfaction with the final outcome.

This decision was driven by the significant variability in the types of materials used and the individual clinical complexity. We recognize this variability as a limitation of our study and recommend future research to investigate the impact of restoration time on patient satisfaction.

The clinical protocol for restorations involved a diagnostic mock-up, which served as a guide for tooth preparation with minimal invasiveness (Figure 4). Based on the type of restorative material and the degree of tooth discoloration, the reduction was carefully controlled to determine the appropriate restoration thickness (Figure 4b and Figure 5).

Lithium disilicate restorations were placed using adhesive cementation techniques. Tooth preparation focused on preserving enamel or, where necessary, superficial dentin. Following the removal of provisional restorations, final restorations were assessed for fit, occlusion, and shade matching (Figure 6; Figure 7). The lithium disilicate crowns were etched for 60 s with 3% to 7% hydrofluoric acid (IPS Ceramic Etching Gel, Ivoclar Vivadent, Schaan, Liechtenstein), thoroughly rinsed, and dried. Any crystalline residue left from the etching process was removed with 36% orthophosphoric acid (Blue Etch, Cerkamed, Stalowa Wola, Poland) for 60 s. The etched surfaces were then treated with silane (Monobond Plus, Ivoclar Vivadent, Schaan, Liechtenstein) for 60 s and dried.

After placing a rubber dam, the teeth were sandblasted with 50 μm aluminum oxide (RØNVIG Dental Mfg. A/S, Daugaard, Denmark) and etched with 36% orthophosphoric acid for 45 s. The surfaces were then rinsed and air-dried before applying adhesive (Adhesive Universal VivaPen, Ivoclar Vivadent, Schaan, Liechtenstein) and air-thinning it to achieve a uniform layer. The final restorations were cemented using dual-cure resin cement (Variolink Esthetic DC, Ivoclar Vivadent, Schaan, Liechtenstein), with both sides of the restoration light-cured for 30 s. Glycerin gel was applied, followed by a final 20 s light cure to ensure full polymerization. Any excess cement was carefully removed, and the restorations were polished after occlusal adjustments.

The selection of composite cement color was carried out using a glycerin-based material to test the compatibility of different light sources, including warm, neutral, and bleach light. This process was carried out to ensure the most natural integration with the surrounding teeth under various lighting conditions. The testing was performed using Variolink Esthetic Try-In material from Ivoclar Vivadent (Schaan, Liechtenstein).

For the zirconia crowns, a similar process was followed. After the try-in, the zirconia crowns were etched with 36% orthophosphoric acid (Blue Etch, Cerkamed, Stalowa Wola, Poland) for 60 s, cleaned, and treated with a ceramic bonding agent (Z-Prime™ Plus, Bisco, Schaumburg, IL, USA) for 60 s. The teeth were then sandblasted, rinsed, and dried before final cementation with a dual-cure self-adhesive resin cement (RelyX™ U200, 3M, Saint Paul, MN, USA). After initial polymerization, excess cement was removed, and the crowns were polished following necessary occlusal adjustments.

For both vital and non-vital teeth, restorations using either lithium disilicate or zirconia were selected based on the individual needs and preferences of each patient. The choice of material was made after discussing the options, outlining the benefits and limitations of lithium disilicate and zirconia, particularly in cases where both materials were suitable (Figure 8, Figure 9, Figure 10 and Figure 11). When there was a clear indication for one material over the other due to the specific circumstances of the case, zirconia or lithium disilicate was chosen by the clinician based on the material’s properties.

For the dyschromia of devitalized teeth, internal bleaching procedures were not performed due to the risk of root resorption or the structural alteration of dental tissues associated with such treatments. Restorative interventions were performed meticulously, with fixed prosthetic restorations featuring margins exclusively placed at the gingival level or subgingivally. The decision not to place veneers on devitalized teeth was based on extensive coronal destruction and the increased risk of tooth fracture associated with the endodontic treatment.

### Statistical Analysis

The statistical analysis was performed using JASPv0.17.1 software. Quantitative variables were expressed as mean ± standard deviation and as median (Quartile1–Quartile3). The comparison between two independent groups was made using the Mann–Whitney U test. The categorical variables were described as number and percentage and the association between the two was made using the Chi2 Test. The results were considered significant for a value of *p* < 0.05.

## 3. Results

The study encompassed a sample of 104 patients undergoing prosthetic treatments, each afflicted with dyschromia in at least one dental unit. Among the participants, 61 were women and 43 were men, with men exhibiting a mean age of 37.721 years, insignificantly lower than the women’s 41.885 year mean age (Mann–Whitney U Test, *p* = 0.093). The study was conducted over the course of one year.

Upon completion of the prosthetic treatments, participants were administered a questionnaire to gauge their satisfaction levels regarding the dental esthetic outcomes. Responses were dichotomized into either satisfaction or dissatisfaction. Dissatisfied participants were further prompted to specify the reasons underlying their discontent, which included chipping, darker final restoration color, and gingival recessions. Among the dissatisfied group, two were men and five were women, and the Yate’s corrected Chi-square test indicated that the proportion of dissatisfied subjects did not significantly vary between genders (*p* = 0.477).

Statistical analyses were conducted to explore potential associations between demographic variables and satisfaction levels with dental esthetics. The Mann–Whitney test revealed that the average age did not significantly differ between subjects satisfied and dissatisfied with dental esthetics (*p* = 0.132). Furthermore, the association between gender and the reason for dissatisfaction was not statistically significant (*p* = 0.094).

These findings suggest a lack of significant age disparity between individuals satisfied and dissatisfied with dental esthetics. Similarly, no statistically significant differences were observed in dissatisfaction levels with dental esthetics between genders, nor in the association between gender and the reason for dissatisfaction.

The outcomes underscore the necessity for further exploration into the multifaceted factors influencing patient satisfaction with dental esthetics post-endodontic and prosthetic treatment of dental dyschromia. A comprehensive understanding of these factors holds the potential to enhance treatment outcomes and elevate the overall quality of patient care.

Table 1 presents the descriptive statistics and comparison of participants’ characteristics by gender. It shows that the mean age for women was 41.89 ± 11.93 years, while for men it was 37.72 ± 9.69 years (*p* = 0.093, Mann–Whitney U test). The median age for women was 42 years (range 32–52), while for men it was 38 years (range 30–45.5). Regarding dissatisfaction, 5 women (8.20%) and 2 men (4.65%) were dissatisfied with their dental esthetic outcomes, with no statistically significant difference between genders (*p* = 0.477, Chi-square test). The reasons for dissatisfaction included chipping (0% in women vs. 4.65% in men, *p* = 0.329), gingival recession (1.64% in women vs. 0% in men, *p* = 0.860), and the final restoration color being darker (5.65% in women vs. 0% in men, *p* = 0.303). 

Table 2 summarizes the patients’ satisfaction results. Of the 104 patients, 97 (93.27%) were satisfied with the dental esthetic outcomes, while 7 (6.73%) were dissatisfied.

## 4. Discussion

In clinical practice, dental treatment following endodontic therapy poses significant challenges [44].

Endodontic success is achieved by following the entire root canal preparation protocol by abundant irrigation with sodium hypochlorite and EDTA, thus achieving canal asepsis. The carious processes in our study were completely removed using a caries detector, ensuring the caries removal was as effective as possible [45,46]. Equally crucial to the success of endodontic treatment is the application of a rubber dam, which effectively isolates the tooth, preventing oral cavity fluids and vapors from contaminating tooth surfaces and root canals. This isolation technique also contributes to the prolonged survival of dental restorations [47].

The tooth’s resistance is enhanced by incorporating glass fiber crown-root devices, which also contributed to achieving a remarkable esthetic appearance, particularly in the context of all-ceramic restorations supported by lithium disilicate [48]. This aspect cannot be taken into account in the case of zirconium oxide restorations due to the disadvantage of its opacity [49]. While rigid metallic post-and-cores were once considered a standard practice in the past, they are no longer a preferred option today due to esthetic concerns and the limited choices they provide [44].

The proper selection of the ceramic block for lithium disilicate restorations is crucial for achieving excellent esthetics, based on the clinician’s registration of the abutment color (ND registration). In another instance, an inadequately opaque lithium disilicate block resulted in noticeable discoloration post fixation, altering the final restoration’s esthetic appearance (Figure 12). Non-vital teeth that have received proper endodontic treatments can remain in the dental arch for unlimited periods due to their favorable survival rates [50,51], if the patients are satisfied with the esthetics of the restorations. They become an integral part of the stomatognathic apparatus, provided that the remaining hard dental tissues allow for a suitable restoration, meeting biological, esthetic, and functional requirements [52].

Endodontically treated teeth exhibit specific characteristics resulting from significant dental hard tissue destruction caused by caries or previous interventions, trophic disorders, and increased vulnerability, particularly at the subgingival level due to the absence of nociceptive sensitivity. The preparation limits for devitalized teeth vary, especially concerning retention. Additionally, these teeth may have dental dyschromias and color changes that, when properly addressed, can lead to long-term benefits in terms of patient satisfaction [3,4].

It is worth discussing the scenario in which, following endodontic treatment failure, retreatments were conducted in all cases, and the compromise of the ferrule effect led to the occurrence of root fractures, particularly in retained metal post-and-core restorations (Figure 13).

The patients’ dissatisfaction stemmed from situations in which, due to various factors (irregular brushing, pregnancy, and periodontal pathology) gingival recessions occurred, exposing discolored teeth (Figure 14).

Another source of dissatisfaction stemmed from the chipping (fractures) that occurred due to aggressive layering when the ceramic thickness exceeding 1.5 mm by the dental technician, aiming to conceal the ZrO ceramic mass for a superior esthetic effect (Figure 15).

In one of the patients dissatisfied with the final color, it was observed that while transparency was lacking, opacity emerged due to the insufficient layering of ceramic material, see Figure 16.

In another instance, the inadequacy of using an insufficiently opaque lithium disilicate block resulted in noticeable discoloration post fixation. The crown exhibited a grayish hue, thereby altering the esthetic appearance of the final lithium disilicate restoration Figure 12.

In this study, in some cases, it was noted that a more aggressive tooth preparation was necessary for the fabrication of fixed prosthodontic restorations. Otherwise, insufficient space for the framework (ZrO framework) and veneering ceramic led to either over-sizing or over-contouring of the restoration, or even to a restoration with poor esthetics [Figure 17].

Although this study exclusively explored concepts related to performing prosthetic restorations, it is necessary to recognize that non-vital tooth bleaching using 30–35% hydrogen peroxide can potentially lead to soft tissue burns [53]. Fortunately, these burns are reversible, and there is no long-term harm to the health of the tissues. In contrast, home-bleaching techniques using low concentrations of hydrogen peroxide may result in systemic effects, such as gastrointestinal mucosal irritation and a burning sensation in the throat and palate [53].

Dental dyschromias are increasingly recognized as a significant concern for patients, directly impacting their perception of dental esthetics. Tooth color has been shown to be the most disturbing factor in dental appearance for 41.25% of patients, with 58.12% able to accurately identify dyschromic teeth in their arches [13,54]. These findings highlight the importance of addressing color-related issues in dental treatment. Our study specifically focused on dyschromias, as tooth color remains one of the most critical and disturbing elements for patients, underscoring the need for tailored solutions that meet both esthetic and functional demands.

Although this study focused on evaluating immediate patient satisfaction, we acknowledge the importance of monitoring satisfaction over the long term, as patient perceptions may evolve based on the durability and functionality of restorations. Future research could explore these aspects to provide a more comprehensive understanding of the long-term impact of such treatments.

## 5. Conclusions

In conclusion, our study demonstrates a notable 93% satisfaction rate among patients who underwent combined endodontic and prosthetic treatments for dental dyschromias.

Understanding the diverse nature of dental dyschromias, including general and locally determined discolorations, is essential for implementing targeted interventions that address the specific needs of each patient. Further research into these factors is essential for ensuring equitable and satisfactory treatment outcomes across diverse patient populations.

## Figures and Tables

**Figure 1 dentistry-13-00044-f001:**
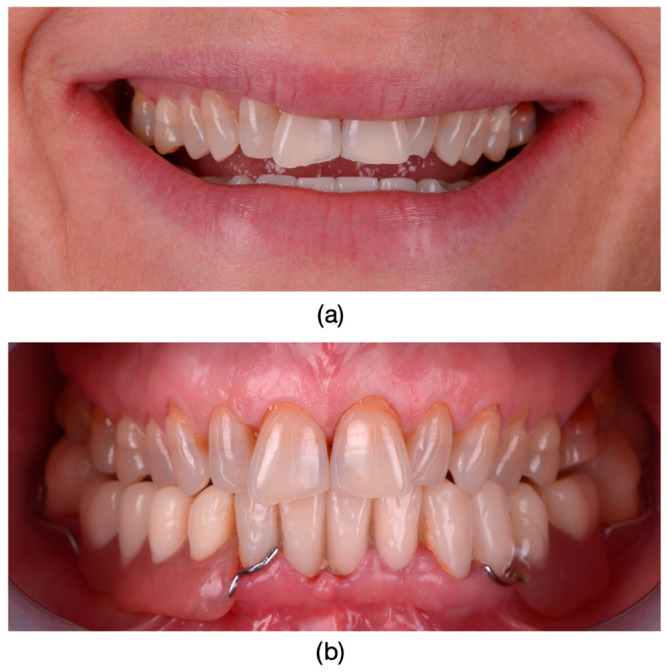
(**a**) The appearance of the patient’s teeth before prosthetic treatment. (**b**) Frontal photograph of the patient prior to treatment initiation, highlighting severe dyschromia associated with tetracycline administration to the mother during the 6th month of pregnancy. The color of the teeth in the upper arch differs from those in the lower arch due to the presence of partially mobile or fixed prosthetic restorations adapted to the remaining teeth or potential prosthetic space in the lower arch.

**Figure 2 dentistry-13-00044-f002:**
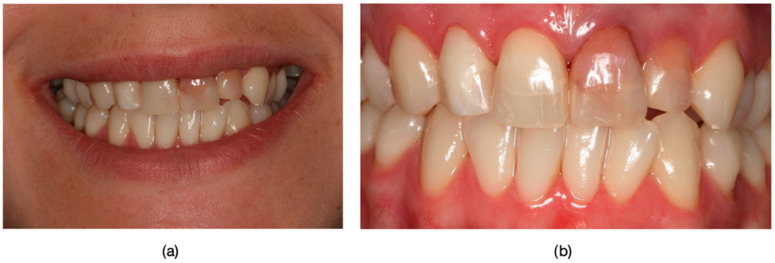
Dyschromia observation in the patient’s smile: (**a**) Patient’s smile from an external perspective, showcasing dental dyschromia and the compromised esthetic appearance. (**b**) Intraoral image of dyschromic teeth: close-up photograph depicting dyschromia in teeth 1.2, 1.1, 2.1, and 2.2 with secondary decays and incorrect endodontical treatments.

**Figure 3 dentistry-13-00044-f003:**
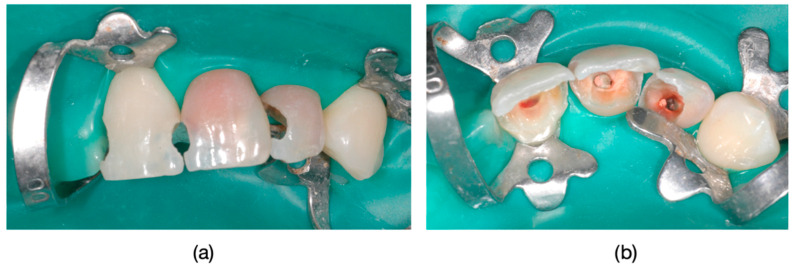
(**a**) Frontal view of dental units 1.1, 2.1, and 2.2 after pre-prosthetic treatment and guided preparation. (**b**) Incisal view of dental units 1.1, 2.1, and 2.2, showing the source of dyschromia originating from the root canal pathways.

**Figure 4 dentistry-13-00044-f004:**
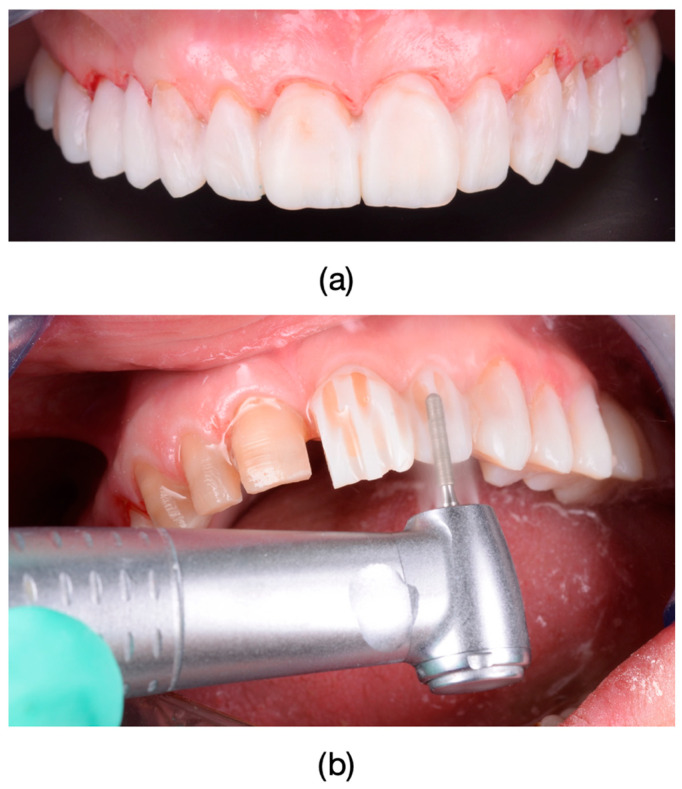
(**a**) To facilitate guided preparation in cases with tetracycline-induced discoloration, a wax-up was created, which was subsequently transferred into the oral cavity as a mock-up. (**b**) Guided preparation using the mock-up.

**Figure 5 dentistry-13-00044-f005:**
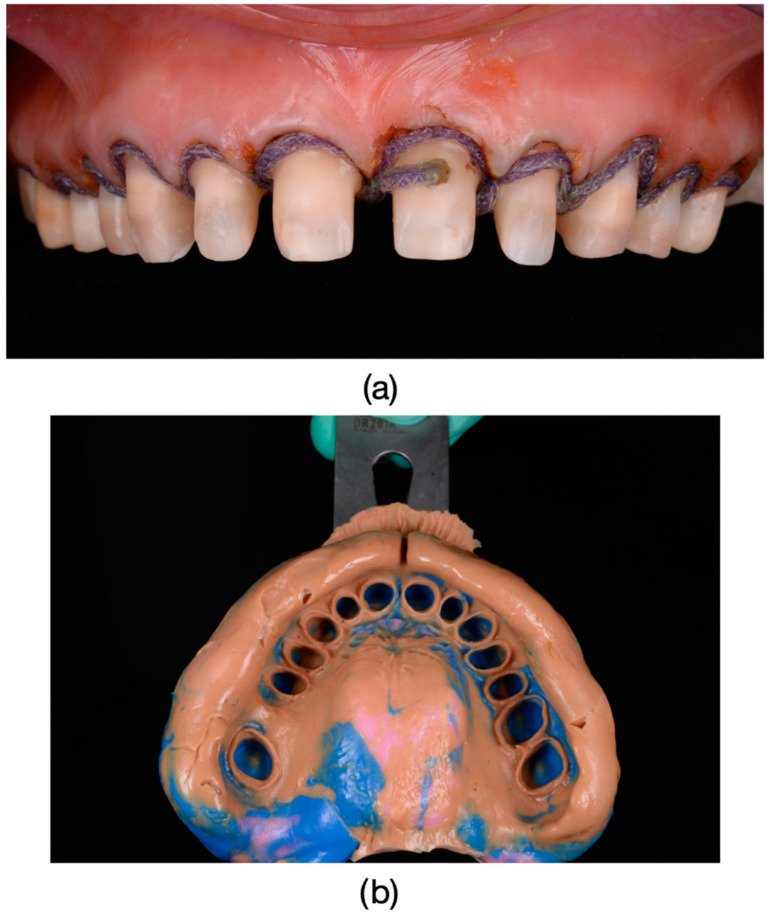
(**a**) Before preparation to accurately reproduce the cervical surface at the level of the prepared chamfer margin, gingival retraction cords (Ultrapak, Ultradent Products, Ultrapack, South Jordan, UT, USA) impregnated with a solution of 25% Aluminum Chloride (ViscostatViscoStat Clear, Ultradent Products, South Jordan, UT, USA) were used. Frontal photograph of the patient after dental preparations, demonstrating dyschromia and nearly symmetrical damage to the tooth structures. (**b**) Impressions were taken using polyvinylsiloxane material (Virtual 380, Ivoclar Vivadent, Schaan, Liechtenstein) using a single-impression double-mixing technique with a standard tray.

**Figure 6 dentistry-13-00044-f006:**
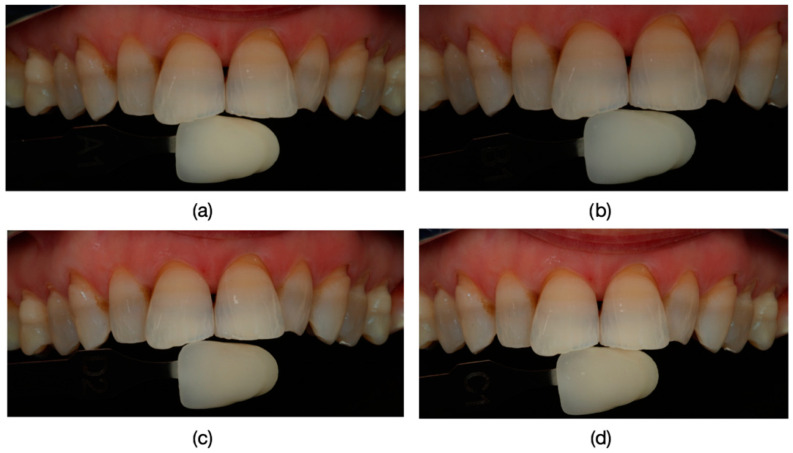
Precise identification of tooth color using Polareys VITA classical shade guide and a polar_eyes cross-polarization filter (Bio-Emulation™, Freiburg im Breisgau, Germany) before tooth preparation. (**a**) Attempted color match A1. (**b**) Attempted color match B1. (**c**) Attempted color match D2. (**d**) Attempted color match C1. Additionally, spectrophotometric methods may also be employed for accurate color assessment.

**Figure 7 dentistry-13-00044-f007:**
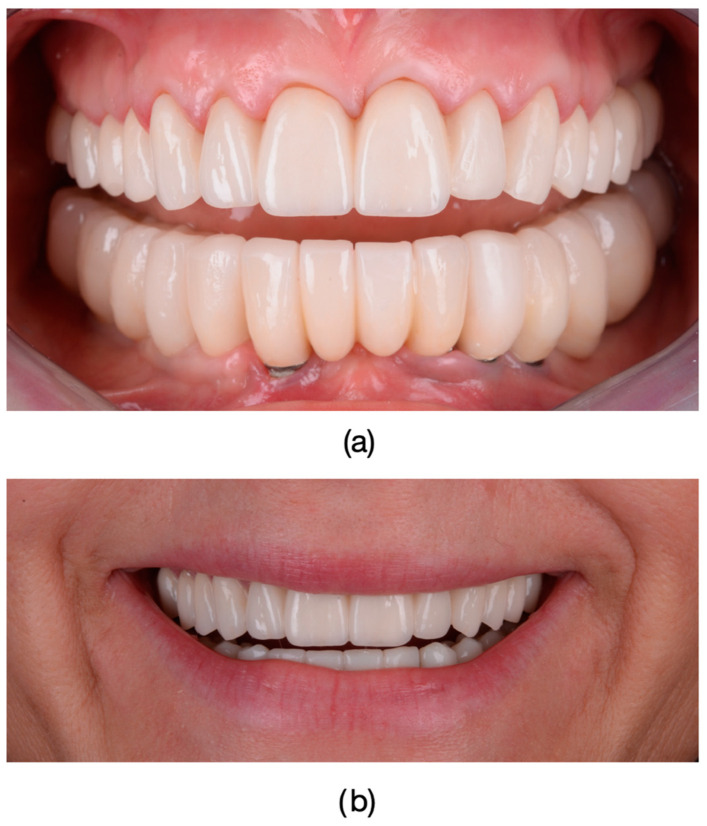
Final fixed restorations: Frontal photograph of the patient after the completion of restorative treatment with full arch-layered zirconia restorations. Final restorations are fixed with a dual cure Self-Adhesive Resin Cement (RelyX™ U200, 3M, Saint Paul, MN, USA), providing the patient with improved dental esthetics and significant reduction in tetracycline-induced dyschromia: (**a**) The appearance of the fixed multi-unit dental restoration in the upper arch. Restorations in the lower arch were performed on dental implants; therefore, no references to the lower arch were made in this study. (**b**) The patient’s smile.

**Figure 8 dentistry-13-00044-f008:**
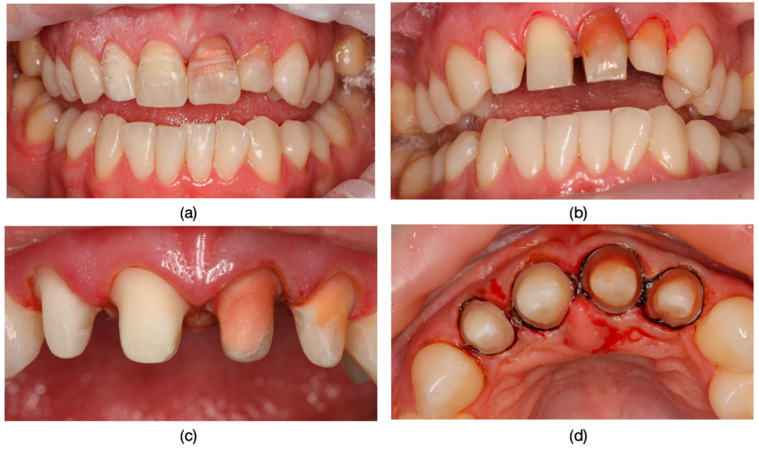
Guided preparation of dental hard tissue: (**a**) Formation of guide grooves to determine the thickness of the prepared dental hard tissue. (**b**) Appearance of dental units after removing 0.2 mm for each shade on the vestibular surfaces to achieve a corresponding final color. (**c**) Appearance of tooth structures after final preparation, revealing asymmetric dyschromia affecting the tooth structures. (**d**) Incisal view of tooth structures with the emphasis on the removed dental hard tissue surface to achieve a correct final color that will blend seamlessly with the overall dental arch context in this regard.

**Figure 9 dentistry-13-00044-f009:**
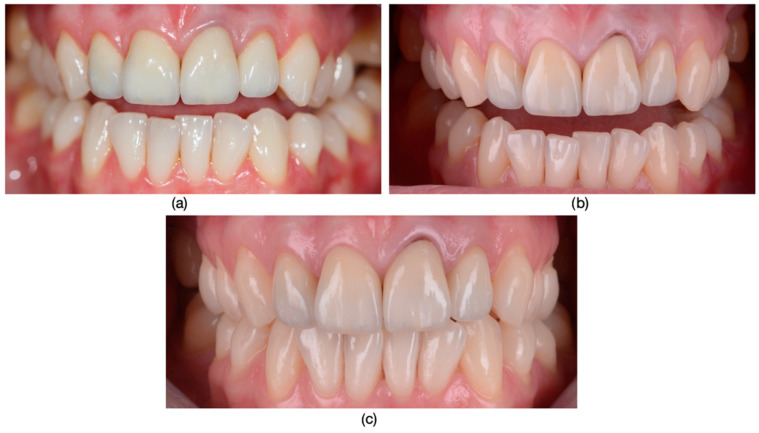
The final appearance and evolution of restorations over time: (**a**) The final esthetic appearance of restorations after fixation. (**b**) The esthetic appearance of restorations after a 3-year recall. During this period, the patient gave birth, and hormonal changes resulted in gingival recession at tooth 2.1, not caused by occlusal overloads as confirmed by occlusal checks with articulation paper of 100μ (Articulating Paper, Bausch, Rochester, NY, USA) both after treatment and at the recall. (**c**) The esthetic appearance of lithium disilicate layered (IPS e.max Press, Ivoclar Vivadent, Schaan, Liechtenstein) after one year from the recall.

**Figure 10 dentistry-13-00044-f010:**
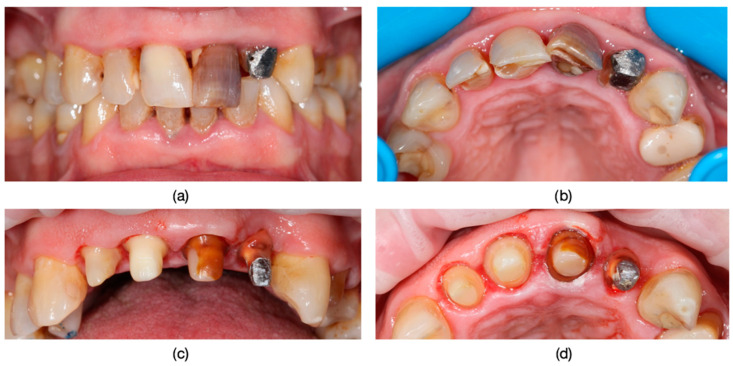
The management of dyschromia and incorrect fillings in dental restorations: (**a**) Dyschromia affecting dental units 2.1 and 2.2 in conjunction with the placement of fixed unidentate prosthetic restorations at 1.1 and 2.1, unaffected by dyschromia but by incorrectly placed fillings. (**b**) Dyschromia and overextended fillings highlighted from an incisal view. (**c**) Preparations performed with frontal visualization, ensuring the proper thickness of dental hard tissue for the uniform esthetic effect of the frontal group. (**d**) Highlighting the prepared chamfer margin along the circumference of the frontal group especially of the teeth with major dyschromia to ensure enough thickness for final restoarations. The color for the final restorations was made in concordance with natural teeth.

**Figure 11 dentistry-13-00044-f011:**
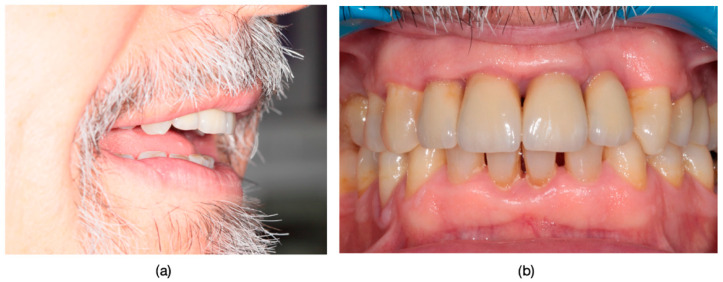
The Esthetic Appearance of Fixed Prosthetic Restorations: (**a**) Esthetic outcome after the cementation of the restorations, shown in the right semiprofile view. (**b**) The intraoral view showcasing the esthetic result of layered zirconia single-crown restorations.

**Figure 12 dentistry-13-00044-f012:**
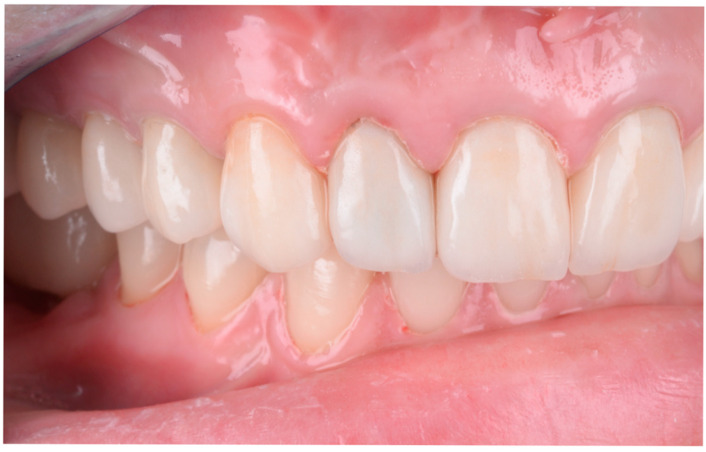
Discoloration on tooth 1.2 post fixation due to the inadequate opacity of the lithium disilicate block.

**Figure 13 dentistry-13-00044-f013:**
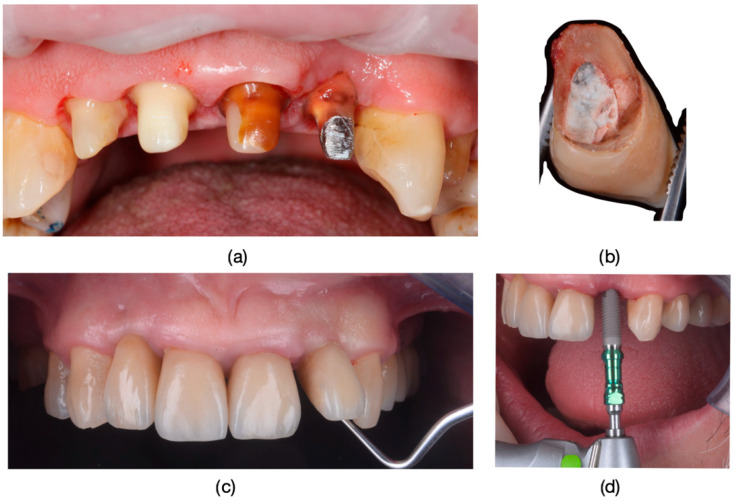
(**a**) The appearance of the teeth before fixation. (**b**) The appearance of tooth 2.2 with a layered zirconia restoration after removal from the oral cavity. (**c**) Mobility caused by the tooth fracture. (**d**) Replacement of the fractured and compromised dental unit through implant treatment.

**Figure 14 dentistry-13-00044-f014:**
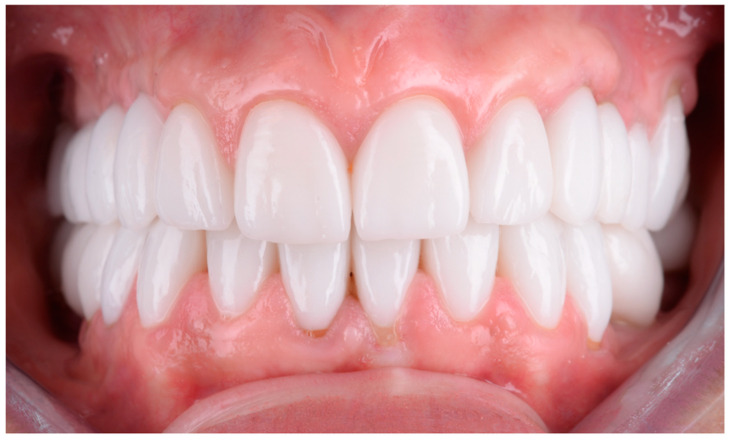
Gingival recession at the level of teeth 3.4, 3.1, and 4.1, attributed to improper brushing techniques.

**Figure 15 dentistry-13-00044-f015:**
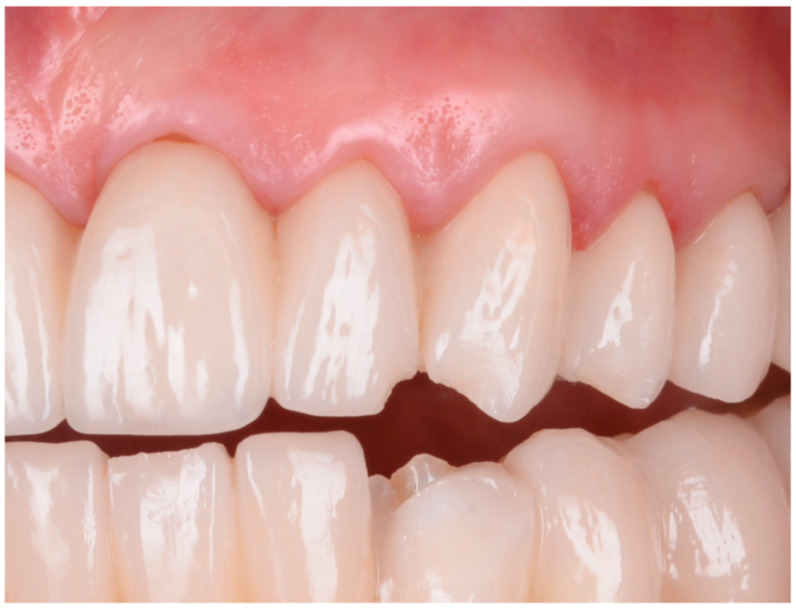
A case of chipping (or fracturing) at the level of a tooth, caused by aggressive layering with an excessive thickness of veneer ceramic by the dental technician. This was carried out to conceal the ZrO ceramic mass, aiming for an improved esthetic effect.

**Figure 16 dentistry-13-00044-f016:**
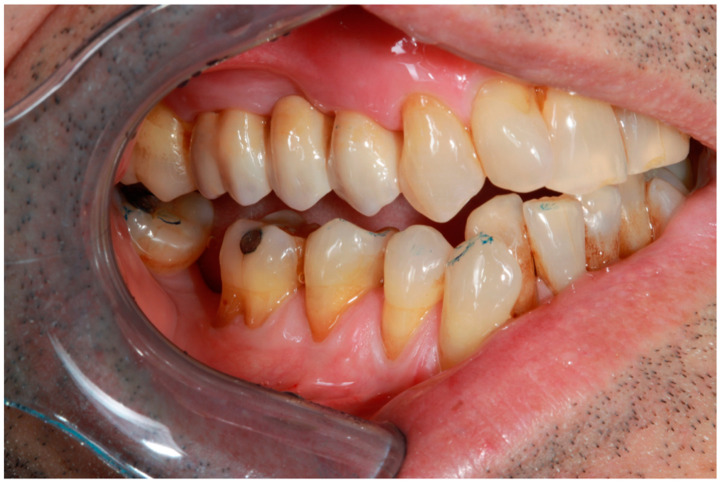
The appearance of the restoration featured exaggerated saturation and tended towards a matte white tone, with opacity becoming evident. The affected dental restorations were 1.4 to 1.6.

**Figure 17 dentistry-13-00044-f017:**
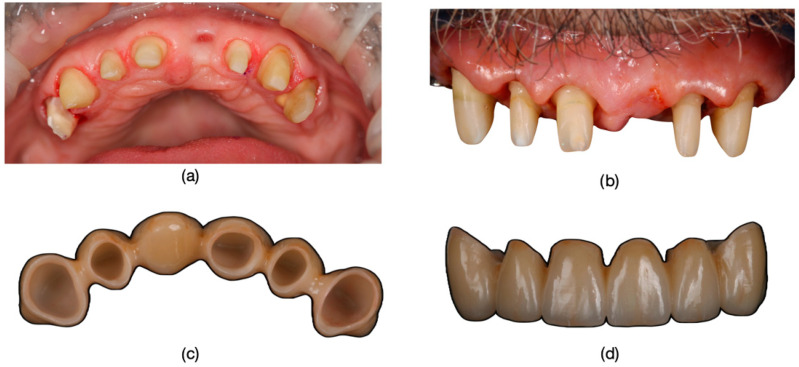
(**a**) For the dyschromic, vital tooth 1.6, endodontic treatment was performed to prevent post-preparation symptoms and facilitate prosthetic restoration. (**b**) Aesthetic view of the dental preparations from the frontal view, showing the parallelism of the preparations. (**c**) View of the restoration from the insertion axis, illustrating the alignment and fit. (**d**) Final restoration appearance from the frontal view, showcasing the completed esthetic result.

**Table 1 dentistry-13-00044-t001:** Descriptive statistics and comparison of participants’ characteristics by gender. (M-W = Mann–Whitney U test; Chi2 = chi-square test).

Variable	Female *N* = 61	Male *N* = 43	*p*
Age as mean ± SD median (Q1–Q2)	41.89 ± 11.93	37.72 ± 9.69	0.093 ^M-W^
	42 (32–52)	38 (30–45.5)	
Dissatisfied *n* (%)	5 (8.20%)	2 (4.65%)	0.477 ^Chi2^
Chipping *n* (%)	0 (0%)	2 (4.65%)	0.329 ^Chi2^
Gingival Recession n (%)	1 (1.64%)	0 (0%)	0.860 ^Chi2^
The Final Restoration Color Was Darker *n* (%)	4 (5.65%)	0 (0%)	0.303 ^Chi2^

**Table 2 dentistry-13-00044-t002:** Patients’ satisfaction results.

Satisfaction	Frequency	Percent
Dissatisfied	7	6.731
Satisfied	97	93.269
Total	104	100

## Data Availability

Data supporting the reported results can be found in the results section.

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
