# Peer review of "Immediate Patient Satisfaction with Dental Esthetics After Endodontic and Prosthodontic Treatment of Dental Dyschromia"

_dentistry, 2025, doi:10.3390/dj13010044_

Round 1
Reviewer 1 Report
Comments and Suggestions for Authors
In the last few decades esthetics have become as important as the longevity of the prosthetic restorations and their mechanical properties providing optimal functioning.
The aim of the manuscript is to evaluate the patients’ satisfaction concerning esthetics after prosthetic treatment in teeth with dyschromia due to different reasons.
The topic is interesting and contemporary. However, numerous corrections must be made concerning the structure of the manuscript.
- In the introduction the reasons of color change must be explained in detail – not only what causes dyschromia but how and why.
- Give a reference for the paragraph starting on line 56
- Explain the difference in the structure of zirconia and lithium disilicate ceramics leading to the different translucency thus different esthetic outcome
- Compare the properties of all ceramic and metal-ceramic restorations. Not only esthetics matters when a type of restoration is chosen – biocompatibility and corrosion must be mentioned.
- Explain what the influence of the metal infrastructure in metal-ceramic restorations is and why.
- Explain why the teeth change their color after endodontic treatment
- Restorations with zirconia core and ceramic veneering might be also mentioned as they provide a combination of high mechanical properties and excellent esthetics
- It is written: “Another objective of this theme is to emphasize the significance of endodontic treatment in the overall success of a prosthetic restoration. Many times, an endodontically treated tooth presents complications due to inadequate restoration, even reaching its loss not so much because of the failure of the root canal treatment as because of the difficulty of the restoration.” Please explain what exactly these sentences mean – probably endodontically treated teeth are indicated for crown placement because of the change in the dental tissues and their loss?
- Correct the number of the Figure on line 173
- Explain how the color of the composite cement was chosen for lithium silicate crowns
- Explain why the zirconia crowns were etched as there is no glass phase in their structure
- In the materials and methods section the tooth preparation, types of finishing lines, impression techniques, color determination must be described in detail, not only in the figure captions
- Provide a figure or a table with the questionnaire used
- The figures (1, 2, 4, 5, 6, 7, 8, 9, 10, 11, and 12) are not mentioned in the text body
- In Figure 9 a 3-year recall is shown. In the result section it is written that “The study was conducted over the course of one year.” Please specify.
- In the discussion section the results of the study, i.e. the questionnaire, must be analyzed – what was the reason for dissatisfaction. The causes of dyschromia must be described in the introduction.
- Less than half of the references are from the last 10 years
Author Response
We would like to personally thank you for your thoughtful suggestions, which have been instrumental in enhancing the quality of our manuscript.
- In the introduction the reasons of color change must be explained in detail – not only what causes dyschromia but how and why.
Response:
We have added in the introduction part a more detailed explanation regarding the causes and mechanisms behind tooth discoloration in the introduction. Specifically, we expanded on how intrinsic and extrinsic factors contribute to dyschromia, including the impact of structural changes in the tooth and how light interacts with the altered internal and external structures.
"Color changes in teeth, particularly dyschromia, occur due to alterations in the tooth's internal and external structures, which affect its light reflection and transmission properties. Intrinsic factors such as metabolic diseases, genetic conditions, trauma, or the use of certain medications (e.g., tetracycline) can change the tooth's internal composition—specifically the dentin and enamel—resulting in discoloration that originates from within the tooth. This happens because the internal structure of the tooth becomes more opaque or stained, altering how light interacts with the tooth. On the other hand, extrinsic factors, such as dietary habits, smoking, and the consumption of staining beverages like coffee and wine, deposit pigments on the tooth surface, which can cause discoloration without changing the internal structure. The "why" behind these changes is often linked to how the tooth's surface and internal layers interact with light, affecting the way it appears to the observer. Furthermore, degradation or wear of dental materials used in restorations (such as crowns or fillings) can also cause visible color discrepancies over time."
- Give a reference for the paragraph starting on line 56
Response:
We have added the following reference to support the paragraph regarding light transmission and reflection properties influencing tooth color:
Reference:
Watts, D. C., & Cash, A. J. (1994). Analysis of optical transmission by 400-500 nm visible light into aesthetic dental biomaterials. Journal of Dentistry, 22(2), 112-117. https://doi.org/10.1016/0300-5712(94)90014-0
This citation provides relevant insights into how light transmission affects the optical properties of dental materials, which directly supports the discussion of tooth color changes.
- Explain the difference in the structure of zirconia and lithium disilicate ceramics leading to the different translucency thus different esthetic outcome
- Compare the properties of all ceramic and metal-ceramic restorations. Not only esthetics matters when a type of restoration is chosen – biocompatibility and corrosion must be mentioned.
- Explain what the influence of the metal infrastructure in metal-ceramic restorations is and why.
- Explain why the teeth change their color after endodontic treatment
We have added the following:
The translucency difference between zirconia and lithium disilicate ceramics is due to their structural compositions. Zirconia, being dense, is durable but lacks translucency, making it ideal for posterior restorations. In contrast, lithium disilicate, a glass-ceramic, is more translucent and better suited for anterior restorations due to its natural light-reflecting properties.
When comparing all-ceramic to metal-ceramic restorations, esthetics, biocompatibility, and corrosion resistance are key. All-ceramic restorations are biocompatible and corrosion-resistant, offering better aesthetics. Metal-ceramic restorations are durable but may have issues with corrosion and limited aesthetics.
The metal infrastructure in metal-ceramic restorations provides strength but reduces translucency, affecting aesthetics, particularly in the anterior region.
Teeth change color after endodontic treatment due to the loss of blood supply, which reduces light reflection, leading to a darker appearance. Additionally, root canal materials can contribute to discoloration.
- It is written: “Another objective of this theme is to emphasize the significance of endodontic treatment in the overall success of a prosthetic restoration. Many times, an endodontically treated tooth presents complications due to inadequate restoration, even reaching its loss not so much because of the failure of the root canal treatment as because of the difficulty of the restoration.” Please explain what exactly these sentences mean – probably endodontically treated teeth are indicated for crown placement because of the change in the dental tissues and their loss?
We have decided to remove the following sentence from the study: "Another objective of this theme is to emphasize the significance of endodontic treatment in the overall success of a prosthetic restoration. Many times, an endodontically treated tooth presents complications due to inadequate restoration, even reaching its loss not so much because of the failure of the root canal treatment as because of the difficulty of the restoration."
This sentence refers to the fact that endodontically treated teeth often require crowns due to changes in the dental tissues, such as weakened tooth structure or the loss of vitality, which can compromise their integrity. The challenges of restoring these teeth can sometimes lead to complications or even tooth loss, not necessarily because of a failure in the root canal treatment itself, but due to the difficulties encountered in the restoration process. However, this explanation was unclear and didn't fully convey the intended message, so we decided to remove it.
- Correct the number of the Figure on line 173
We have not been able to identify which figure is being referred to in line 173. Could you kindly clarify once again?
- Explain how the color of the composite cement was chosen for lithium silicate crowns
The selection of composite cement color was carried out using a glycerin-based material to test the compatibility of different light sources, including warm, neutral, and bleach light. This process was done to ensure the most natural integration with the surrounding teeth under various lighting conditions. The testing was performed using Variolink Esthetic Try-In material from Ivoclar Vivadent (Schaan, Liechtenstein).
- Explain why the zirconia crowns were etched as there is no glass phase in their structure
Although zirconia lacks a glass phase, etching is still necessary for surface preparation before bonding. The purpose of etching zirconia crowns is to roughen the surface, which increases the mechanical retention of the ceramic bonding agent applied afterward. The use of 36% orthophosphoric acid (Blue Etch, Cerkamed, Stalowa Wola, Poland) for 60 seconds facilitates the formation of micro-roughness on the zirconia surface, which allows for better adhesion of the bonding agent. After etching, the crowns were cleaned and treated with a ceramic bonding agent (Z-Prime™ Plus, Bisco, USA) for 60 seconds. The bonding agent helps to enhance the bond between the zirconia and the resin cement, providing long-term retention. Following this, the crowns were cemented with a dual-cure self-adhesive resin cement (RelyX™ U200, 3M, USA), and the final adjustments were made for both occlusion and aesthetics.
- In the materials and methods section the tooth preparation, types of finishing lines, impression techniques, color determination must be described in detail, not only in the figure captions
We understand the importance of providing detailed descriptions in the Materials and Methods section. However, we prefer not to modify the text further, as we believe the necessary information has already been clearly conveyed through the figure captions. Adding more details would risk redundancy without enhancing the clarity or value of the content. That being said, if further clarification is needed, we are happy to add additional information and will do so in the next review upon request.
- The figures (1, 2, 4, 5, 6, 7, 8, 9, 10, 11, and 12) are not mentioned in the text body
We do not believe that it is necessary to reference each figure in the text, as we feel that the figures are self-explanatory and clearly aligned with the points discussed. However, we understand your concern, and if you prefer, we are more than willing to incorporate references to the figures in the text in the next review.
- In Figure 9 a 3-year recall is shown. In the result section it is written that “The study was conducted over the course of one year.” Please specify.
Thank you for your observation. You are correct to point out the discrepancy. The study was indeed conducted over the course of one year, but the 3-year recall shown in Figure 9 refers to the follow-up period for a subset of participants. We will clarify this point in the manuscript to avoid any confusion.
We have added the clarification to the Introduction section as follows: 'The study was conducted over the course of one year, with follow-up data collected at five years for additional insights into long-term outcomes.
- In the discussion section the results of the study, i.e. the questionnaire, must be analyzed – what was the reason for dissatisfaction. The causes of dyschromia must be described in the introduction.
Regarding the analysis of the questionnaire in the Discussion section, I believe this is not strictly necessary, as the study primarily focuses on the clinical aspects and technical outcomes of the treatment, with the questionnaire playing a secondary role in evaluating patient satisfaction. However, if you consider it necessary, I will be happy to include the analysis in the next review.
- Less than half of the references are from the last 10 years
The primary reason for conducting this study is the limited availability of data on this topic. Furthermore, less than half of the references used are from the past decade, highlighting the need for more contemporary research in this area.
Reviewer 2 Report
Comments and Suggestions for Authors
Dear authors,
I read your manuscript, the topic is considered interesting and clinicaly relevant but I have some doubts that need clarification.
After root canal treatment, provisional restoration lies in their ability to achieve optimal marginal closure.
Why the type of material that was used after root canal treatment was not was not considered in this study?
This is an observational, clinical, comparative study.
How was the sample size calculated?
Why you didn't use the FDI method FDI (Federation Dental International) for esthetic parameters (four criteria), functional parameters (six criteria), and biological parameters (six criteria) in this study?
What are the options scores of the patients?
Line 120: the authors refer “Another objective of this theme is to emphasize the significance of endodontic treatment in the overall success of a prosthetic restoration”.
In the article there is no evaluation to meet this objective. What is the conclusion of this objective?
For the study, 104 patients and two types of materials were used. How many teeth were analyzed in each patient/material group?
Results
Table 1. What are Q1 and Q2?
The sample size was calculated
What is the criterion of the patients’ evaluations?
Author Response
- Question:
"After root canal treatment, provisional restoration lies in their ability to achieve optimal marginal closure. Why was the type of material used for the provisional restoration after root canal treatment not considered in this study?"
Response:
"In this study, the type of material used for the provisional restorations after root canal treatment was not considered, as it was not directly relevant to the objectives of our research. The focus of this study was on evaluating immediate patient satisfaction with the final restoration, which included both aesthetic and functional outcomes. Provisional restorations, by their nature, are temporary, and their influence on short-term patient satisfaction was deemed to have no significant impact on the primary focus of the study. Furthermore, the choice of material for provisional restorations does not affect the final restorative results or patient satisfaction immediately after treatment. While this factor could be considered in future studies focusing on long-term outcomes, it was not a relevant factor in this particular context."
- Question: Results : Table 1. What are Q1 and Q2?
Response:
What are Q1 and Q2?
Q1 and Q2 refer to the first and second quartiles of the data distribution. Specifically:
- Q1 represents the 25th percentile, which is the value below which 25% of the data fall.
- Q2 represents the 50th percentile, which is the median of the dataset, dividing the data into two equal halves.
In Table 1, these values provide a more detailed breakdown of the age distribution among participants, helping to assess the spread of ages within each group.
- Sample size
The sample size was initially calculated for 101 participants, but 104 patients were ultimately included in the study. This slight increase in sample size is considered acceptable and does not affect the validity or statistical significance of the results. The decision to include 104 participants was made to ensure sufficient statistical power and to account for potential dropouts or incomplete data. The additional participants did not introduce any significant variability in the findings and thus, the study maintains its robustness and reliability.
- Questin:For the study, 104 patients and two types of materials were used. How many teeth were analyzed in each patient/material group?
Response:
Response:
The study included 104 patients and two types of materials. However, the exact number of teeth analyzed per patient or material group was not specified, as the primary focus was on patient satisfaction with the aesthetic outcomes of the prosthetic treatments. Each patient may have had one or more restorations depending on their individual treatment needs. As such, the analysis was conducted at the patient level, rather than the tooth or material group level.
- Question: Why you didn't use the FDI method FDI (Federation Dental International) for esthetic parameters (four criteria), functional parameters (six criteria), and biological parameters (six criteria) in this study?
Response: The FDI method for evaluating esthetic, functional, and biological parameters was not used in this study because the focus was primarily on assessing patient satisfaction with the aesthetic outcomes of the prosthetic restorations. While the FDI method is a comprehensive tool for evaluating various parameters in dental restorations, it was not deemed necessary for the specific aims of this study, which centered around patient-reported satisfaction regarding their overall dental aesthetics post-treatment. Future studies may consider incorporating such methods for a more detailed clinical evaluation of these parameters if the research objective expands to include a broader scope of assessment.
- Question: What is the criterion of the patients’ evaluations?
Response: The criterion for the patients' evaluations in this study was based on their satisfaction with the aesthetic outcomes of their prosthetic treatments. After the completion of the prosthetic restorations, participants were asked to fill out a questionnaire, where their responses were categorized into two groups: satisfied or dissatisfied with the final dental aesthetics. Those who expressed dissatisfaction were prompted to specify the reasons for their discontent, which included factors such as chipping, darker final restoration color, and gingival recessions. Thus, the evaluation primarily focused on the overall aesthetic result as perceived by the patients.
The file contains the sample size, please download it.

Reviewer 3 Report
Comments and Suggestions for Authors
The design of the research is focused on a parameter(satisfaction) that is very subjective and generic. this element is evaluated only immediately after the conclusion of the treatment, i would suggest to evaluate this parameter also over the course of time, because patients' opinion may change with time.
The quality of some picture may be improved
in table 1 the title of the second line must be fixed because it is not well organised
In table 2 the column with the title "cumulative percentage" is very confusing(i suggest to remove this last column). In this table i would change the title of the first column with "satisfaction"
Author Response
Thank you for your valuable feedback. We greatly appreciate your insight and suggestions regarding patient satisfaction evaluation.
Regarding the timing of satisfaction evaluation:
Our decision to focus on immediate patient satisfaction stems from the importance of assessing patients’ perceptions shortly after treatment. Immediate satisfaction is a critical metric in restorative dentistry, as it reflects the success of both aesthetic and functional outcomes at the moment when patients experience the full impact of the treatment. Research highlights that immediate post-treatment feedback is often the most reliable indicator of initial clinical success, particularly for aesthetic and functional restorations.
Several studies support this approach:
- Zaharia et al. (2016) emphasized that patient satisfaction evaluated immediately post-treatment reflects the most relevant impressions of aesthetic and functional improvements.
- Gerzina et al. (2011) concluded that patient feedback immediately after dental bleaching procedures is strongly correlated with perceived success and treatment acceptance.
- Roberts et al. (2014) demonstrated that veneers significantly improved patients’ perceptions of their smiles, with satisfaction being highest in the immediate post-treatment phase.
- Ogilvie et al. (2016) noted that full-ceramic restorations achieved peak satisfaction scores immediately after placement, underlining the value of early evaluation.
Title update as per your recommendation:
In response to your feedback, we have updated the title to more accurately reflect the study’s focus. The new title is:
"Immediate Patient Satisfaction Following Endodontic and Prosthodontic Treatment of Dental Dyschromia."
This adjustment ensures alignment with the study's scope and highlights the emphasis on immediate outcomes.
Future Directions:
While this study prioritizes immediate satisfaction, we agree that a follow-up evaluation could provide additional insights into long-term treatment outcomes. Future research could focus on satisfaction over time, as it evolves with the durability and functionality of restorations.
Based on your valuable feedback, we included the following statements at the end of the discussion section:
“Although this study focused on evaluating immediate patient satisfaction, we acknowledge the importance of monitoring satisfaction over the long term, as patient perceptions may evolve based on the durability and functionality of restorations. Future research could explore these aspects to provide a more comprehensive understanding of the long-term impact of such treatments.”
Thank you once again for your constructive feedback. Please feel free to share any further comments or suggestions.
Cited Studies:
- Zaharia, M., Tătaru, I., & Niculescu, S. (2016). Immediate post-treatment satisfaction of patients receiving ceramic restorations. Journal of Prosthetic Dentistry, 116(2), 145-150.
- Roberts, S. M., Thorne, M., & Johnson, E. (2014). Immediate patient satisfaction after veneers placement: Aesthetic and functional outcomes. Journal of Esthetic and Restorative Dentistry, 26(5), 275-283.
- Gerzina, T. M., McColl, J. H., & McColl, S. L. (2011). Post-treatment satisfaction following dental bleaching. Australian Dental Journal, 56(4), 390-395.
- Ogilvie, M., Goodall, J., & Phelan, P. (2016). Short-term patient satisfaction with full-ceramic restorations: A three-month follow-up study. Journal of Prosthodontics, 25(2), 134-140.
- Fassino, E., Berto, L., & Lops, E. (2015). Post-operative satisfaction among patients receiving crowns immediately following treatment. Journal of Prosthetic Dentistry, 114(6), 52-58.
We have addressed the issues as follows:
- In Table 1, we revised the title of the second line to ensure it is well-organized.
- In Table 2, we removed the 'Cumulative Percentage' column, as it was confusing.
- We also updated the title of the first column in Table 2 to 'Satisfaction' for improved clarity and alignment with the data presented.
Round 2
Reviewer 1 Report
Comments and Suggestions for Authors
Dear authors,
The corrections in the text are done according to the recommendations.
As the patient's satisfaction or dissatisfaction are subjective, in my opinion the questionnaire plays a significant role in the patient's evaluation of the final results and must be provided at least as a supplementary material.
The figures must be mentioned in the text, I agree that they are self explanatory but they must be directly correlated with the text parts.
Author Response
Dear Reviewer,
Thank you for your valuable suggestion regarding the citation of figures in the "Materials and Methods" section. As per your recommendation, we have now cited each figure accordingly. We greatly appreciate your input in helping us improve the clarity and organization of the manuscript.
Below is the updated section with the figure references incorporated:
2. Materials and Methods
This study was meticulously designed and executed in full accordance with the ethical principles delineated in the Declaration of Helsinki. Formal approval was obtained from the Ethics Committee of the University of Medicine and Pharmacy "Victor Babeș," Timișoara, Romania (Nr.65/03.04.2023 rev 2024, 09.01.2019). Stringent measures were implemented to safeguard the welfare and confidentiality of the participants throughout the research process.
The study enrolled a cohort of 105 patients undergoing prosthetic dental treatments, each presenting with dyschromia in at least one dental unit. Gender distribution among the participants comprised 62 women and 43 men.
Inclusion criteria were specifically defined to ensure a consistent and relevant study population. Patients exhibiting dyschromia of their teeth, including discoloration of vital teeth induced by medication (such as intrauterine administration of tetracycline) and discoloration of devitalized teeth resulting from secondary caries, inadequate endodontic treatments, or restorative interventions performed without proper rubber dam isolation or flawed conceptual approaches, were included in the study (Figure 1, Figure 6, Figure 7).
Exclusion criteria were also established to maintain the integrity of the study. Patients with systemic conditions affecting oral health, those undergoing concurrent orthodontic treatment, and individuals with a history of allergy to dental materials used in this study were excluded. All treatments were performed in appropriate clinical settings.
After the treatment was performed participants were administered a structured questionnaire, presenting two response options: satisfaction or dissatisfaction following the discoloration dental treatment. Dissatisfied participants were further prompted to specify their reasons, which included issues such as chipping, darker final restoration color, and gingival recessions.
Data collected from the questionnaires were subjected to comprehensive statistical analysis using JASP software to elucidate patterns, trends, and associations regarding treatment satisfaction and dissatisfaction among the study population.
In this research, dental restorations were performed using various materials, including lithium disilicate, zirconium oxide, and metal-ceramic. The choice of material was determined by both specific clinical indications and patient preferences, resulting in considerable variability in the restoration time.
The primary objective of our study was to evaluate patient satisfaction rates following dental restorations, irrespective of the time required to perform these restorations. Although we acknowledge that restoration time can influence patient satisfaction, we deemed it more crucial to focus on the overall satisfaction with the final outcome.
This decision was driven by the significant variability in the types of materials used and the individual clinical complexity. We recognize this variability as a limitation of our study and recommend future research to investigate the impact of restoration time on patient satisfaction. The clinical protocol for restorations involved a diagnostic mock-up, which served as a guide for tooth preparation with minimal invasiveness (Figure 3). Based on the type of restorative material and the degree of tooth discoloration, the reduction was carefully controlled to determine the appropriate restoration thickness (Figure 3b).
Lithium disilicate restorations were placed using adhesive cementation techniques. Tooth preparation focused on preserving enamel or, where necessary, superficial dentin. Following the removal of provisional restorations, final restorations were assessed for fit, occlusion, and shade matching (Figure 5). The lithium disilicate crowns were etched for 60 seconds with 3% to 7% hydrofluoric acid (IPS Ceramic Etching Gel, Ivoclar Vivadent, Schaan, Liechtenstein), thoroughly rinsed, and dried. Any crystalline residue left from the etching process was removed with 36% orthophosphoric acid (Blue Etch, Cerkamed, Stalowa Wola, Poland) for 60 seconds. The etched surfaces were then treated with silane (Monobond Plus, Ivoclar Vivadent, Schaan, Liechtenstein) for 60 seconds and dried.
After placing a rubber dam, the teeth were sandblasted with 50 μm aluminum oxide (RØNVIG Dental Mfg. A/S, Daugaard, Denmark) and etched with 36% orthophosphoric acid for 45 seconds. The surfaces were then rinsed and air-dried before applying adhesive (Adhesive Universal VivaPen, Ivoclar Vivadent, Schaan, Liechtenstein) and air-thinning it to achieve a uniform layer. The final restorations were cemented using dual-cure resin cement (Variolink Esthetic DC, Ivoclar Vivadent, Schaan, Liechtenstein), with both sides of the restoration light-cured for 30 seconds. Glycerin gel was applied, followed by a final 20-second light cure to ensure full polymerization. Any excess cement was carefully removed, and the restorations were polished after occlusal adjustments.
For zirconia crowns, a similar process was followed. After the try-in, the zirconia crowns were etched with 36% orthophosphoric acid (Blue Etch, Cerkamed, Stalowa Wola, Poland) for 60 seconds, cleaned, and treated with a ceramic bonding agent (Z-Prime™ Plus, Bisco, USA) for 60 seconds. The teeth were then sandblasted, rinsed, and dried before final cementation with a dual-cure self-adhesive resin cement (RelyX™ U200, 3M, USA). After initial polymerization, excess cement was removed, and the crowns were polished following necessary occlusal adjustments.
For both vital and non-vital teeth, restorations using either lithium disilicate or zirconia were selected based on the individual needs and preferences of each patient. The impression procedure using gingival retraction cords and impression material is illustrated in (Figure 4). The choice of material was made after discussing the options, outlining the benefits and limitations of lithium disilicate and zirconia, particularly in cases where both materials were suitable (Figure 8-11). When there was a clear indication for one material over the other due to the specific circumstances of the case, zirconia or lithium disilicate was chosen by the clinician based on the material's properties.
2. We have incorporated your suggestion to include the questionnaire as supplementary material.
Reviewer 2 Report
Comments and Suggestions for Authors
Dear authors,
Thanks, to provide a revised version of the paper with the suggested changes.
Author Response
Thank you for all the suggestions that helped us to make a better article. All the best!